# Is the Letter 't' in the Word *'gourmet'*? Disruption in Task-Evoked Connectivity Networks in Adults with Impaired Literacy Skills

**Kulpreet Cheema** [1,*] **, William E. Hodgetts** [2,3] **and Jacqueline Cummine** [1,2]

1 Department of Neuroscience and Mental Health, Faculty of Medicine and Dentistry, University of Alberta, Edmonton, AB T6G 2H7, Canada; jcummine@ualberta.ca
2 Department of Communication Sciences and Disorders, Faculty of Rehabilitation Medicine, University of Alberta, Edmonton, AB T6G 2G4, Canada; hodgetts@ualberta.ca
3 Institute for Reconstructive Sciences in Medicine, Covenant Health, Edmonton, AB T5R 4H5, Canada
* Correspondence: kulpreet@ualberta.ca

**Abstract:** Much work has been done to characterize domain-specific brain networks associated with reading, but very little work has been done with respect to spelling. Our aim was to characterize domain-specific spelling networks (SpNs) and domain-general resting state networks (RSNs) in adults with and without literacy impairments. Skilled and impaired adults were recruited from the University of Alberta. Participants completed three conditions of an in-scanner spelling task called a letter probe task (LPT). We found highly connected SpNs for both groups of individuals, albeit comparatively more connections for skilled (50) vs. impaired (43) readers. Notably, the SpNs did not correlate with spelling behaviour for either group. We also found relationships between SpNs and RSNs for both groups of individuals, this time with comparatively fewer connections for skilled (36) vs. impaired (53) readers. Finally, the RSNs did predict spelling performance in a limited manner for the skilled readers. These results advance our understanding of brain networks associated with spelling and add to the growing body of literature that describes the important and intricate connections between domain-specific networks and domain-general networks (i.e., resting states) in individuals with and without developmental disorders.

**Keywords:** reading impairment; task connectivity; brain–behaviour relationship; resting-state networks; spelling network; inferior frontal gyrus





## 1. Introduction

The field of neuroscience of literacy (e.g., reading and spelling) has moved beyond descriptions of differences in mean brain activation to instead consider the dynamic neural networks that are associated with such complex and high-order task processes [1,2]. This shift in our approach to studying literacy has resulted in new perspectives that shed light on literacy acquisition [3,4], refinement [5–7], and break down [8,9]. While such advances have been substantial in the reading domain, the spelling domain remains comparatively less well understood. The current paper aimed to address this gap in understanding of literacy, which requires that we: (1) characterized the domain-specific spelling network (SpN) of skilled and literacy-impaired individuals, (2) determined the extent to which the domain-specific network is related to domain-general (e.g., attention-specific) resting state networks (RSNs), (3) quantified how the domain-specific spelling network is related to spelling performance, and, finally, (4) assessed the extent to which domain-general brain networks are related to spelling behavior.

### 1.1. Domain-Specific Spelling Network

Literacy is the ability to read and write to communicate with, and understand, the world around us. Spelling is a form of written language production that requires

the knowledge and coordination of various linguistic functions including phonology (i.e., units of sound), morphology (i.e., units of meaning), and orthography (i.e., units of print) [10]. Compared to reading, comparatively little has been done to advance the neuroanatomical models of written language production (e.g., spelling and/or writing) [11–14]. With respect to skilled adults, studies on spelling have provided some insight into the brain regions and corresponding processes that are involved in the generation and retrieval of spelling representations [14–18]. The identified brain regions include the supramarginal gyrus (SMG) and superior temporal gyrus (STG) (i.e., sound; [19]), the angular gyrus (AG) and middle temporal gyrus (MTG) (i.e., meaning; [15,18], and the fusiform gyrus (FFG) and inferior temporal gyrus (ITG) (i.e., print; [15–17]). Interestingly, while the brain regions associated with spelling substantially overlap with those identified for reading, there is evidence that brain function/structure differentially predicts reading vs. spelling outcomes [20,21]. Therefore, brain–behaviour relationships associated with spelling need to be examined explicitly.

Behavioural and neuroimaging studies on individuals with spelling impairments have also provided additional information on spelling processes. In the behaviour domain, individuals with spelling disorders have been found to commit multiple kinds of spelling errors, including phonological, orthographic, and morphological spelling errors [22–24], consistent with the much reported heterogeneous profiles of reading behaviour. Brain imaging studies performed on participants with the combined profile of reading and spelling deficits (i.e. dyslexia) and isolated spelling impairments have also shed light on the brain mechanisms involved in spelling [4,25]. For example, in a study comparing children with combined reading and spelling deficits and isolated spelling disorder, a widespread decrease of brain activation was observed in children with the combined profile [26]. These brain regions included both the dorsal and ventral areas of the reading network. In another study, researchers reported an increased activation in the right hemisphere (e.g., supramarginal gyrus—sound) compared to skilled individuals and under-activation in left inferior temporal regions (e.g., fusiform gyrus—print; [9,27]. Such findings have been taken as evidence for compensatory and/or impaired sound and print processing, respectively. Similarly, in a study that included participants with impaired spelling abilities following stroke, the authors reported that atrophy in the left supramarginal gyrus (sound) and the inferior frontal gyrus (IFG; production) was associated with non-word spelling errors, while atrophy of the inferior temporal region was related to real word spelling errors [19]. Together, these studies provided some preliminary information about the brain areas involved in skilled and impaired spelling processes. The extent to which these previous findings also generalize to adults with literacy impairments has yet to be established, despite the overwhelming evidence of lifelong spelling struggles in adults with developmental literacy impairments (e.g., dyslexia). Notably, Riddick and colleagues (1999) stated that "spelling problems constitute the most prominent and persistent difficulty which they (adults with dyslexia) encounter" (p. 228) [28]. We address this gap in the current paper by describing the dynamic brain networks related to spelling in adults with (and without) a history of literacy impairments.

*1.2. From Brain Activation to Brain Connectivity*

Recently, researchers have gone beyond a description of the mean activation of particular brain areas associated with specific tasks to testing how multiple brain regions interact with one another (i.e., connectivity) to support the completion of various tasks. While there have been no such investigations with spelling tasks to date, several informative reading-based studies have been conducted. Specifically, there is mounting evidence for reduced connectivity from occipitotemporal regions involved in print processing (e.g., fusiform gyrus) to regions associated with sound processing (i.e., inferior parietal lobule) and production and/or motor representations (inferior frontal gyrus and precentral gyri; [3,6,9]) in individuals with poor reading abilities. An increased connectivity between the production (i.e., left inferior frontal gyrus) and sound (i.e., the caudate) based regions was

also reported by [3], and it was described as a compensatory strategy of sounding out words. Together, these studies provided evidence for altered functional connectivity among reading-specific brain regions in individuals with poor literacy skills. However, we do not yet know whether similar hyper/hypo connections are also characteristic of spelling tasks. More importantly, the extent to which the strength of these brain connections relates to behavioural performance for skilled and impaired individuals has yet to be determined.

### 1.3. Domain-General Resting State Networks (RSN)

Beyond the domain-specific brain regions known to contribute to spelling, it is also important to consider how domain-general skills, like attention and working memory, relate to spelling performance. For example, while spelling to dictation, one not only requires the integration of sound and letter information but also to store and access these representations in our working memory in order to spell out the words. These domain-general skills are robustly represented in the brain in the form of resting-state networks [2,29,30]. Drawing from the reading literature, there is evidence that domain-specific networks and domain-general networks are coupled in meaningful ways. For example, reading related brain regions and networks are related to domain-general resting state networks (RSNs), including the default mode network (DMN) [6], attentional networks [31], and salience networks [29,32]. Of particular importance to the current work, Bailey and colleagues (2018) found that regions in the dorsal attention network (DAN) contributed the most to the reading-related activation [33], and Vogel and colleagues (2012) reported that the visual word form area (VWFA) had stronger connections to regions in the dorsal attention network over and above the reading-related regions [34]. Similarly, the DMN is consistently reported to be negatively connected to various reading-related regions [3,5,6]. For example, the authors of [35] recently argued that the intrinsic connectivity of the precuneus cortex, a key region in the DMN, relates to an increase in focus during a reading comprehension task, thereby contributing positively to the reading process. Beyond the DMN and attentional networks, the salience network, an executive control network involved in monitoring behavioural goals in response to salient stimuli (e.g., adjusting attention and error detection), has been reported to be strongly connected to scholastic performance [31], including reading (see also [30] for a discussion of task-based connectivity and reduced salience connectivity for children with reading impairments). Understanding the extent to which such findings generalize to spelling-related networks was one of the goals of the current work.

Research into the relationships between resting-state and task-based networks associated with spelling performance has been negligible, particularly for adults with poor literacy skills. There is some evidence from the reading literature that points to an increased within-network connectivity among DMN regions [3] for impaired readers, possibly signifying compensatory mechanisms and/or strategies for these individuals. Additionally, noteworthy are reports of reduced connectivity between DMN regions and the fusiform gyrus in children and adults with reading impairments when compared to skilled readers [31,36]. While this preliminary body of evidence for intrinsic connections in skilled vs. impaired readers is important, we cannot assume that such findings generalize to spelling behaviour and spelling networks or to an adult population. To better understand how the inherent functional architecture contributes to the dynamic spelling processes, we need to provide more specificity with respect to the connectivity associated with spelling-related brain regions and, ultimately, connect such networks to spelling behaviour.

### 1.4. Goals of the Current Work

Given the body of reviewed work, the overall goal of this paper was to characterize the brain networks for spelling processes in adults with and without literacy impairments. To do this, we first described the domain-specific spelling networks of skilled and literacy-impaired individuals. Next, we examined how the domain-specific network is related to domain-general (e.g., attention-specific) brain networks in each group. Third, we quantified

how the domain-specific spelling network is related to spelling performance. Finally, we assessed the extent to which domain-general brain networks are related to spelling behaviour.

## 2. Materials and Methods

### 2.1. Participants

Participants (N = 34) were recruited via advertisements through email, postings in online forums, and posters. All individuals took part in both the behavioural and imaging aspects of the study. Of these, 14 adults were classified as having reading impairment (referred to as the impaired group) (4 males; mean age = 24.36 years), and 19 were skilled readers (referred to as the skilled group) (5 males; mean age = 21.58 years). The inclusion criteria for the skilled group consisted of English as the native or primary language, normal or corrected to normal vision, no contraindications to go in the MRI, and age-appropriate scores on reading, spelling, and IQ measures. Inclusion criteria for the impaired group consisted of English as the native or primary language, normal or corrected to normal vision, no contraindications to go in the MRI, and age-appropriate score on the nonverbal IQ test. In addition, participants in the impaired group (1) self-identified as having a literacy impairment and (2) scored > 1.5 SD below the skilled group on at least one of the standardized reading tasks (tests described below; [37]). Exclusion criteria for both groups included a history of any hearing or vision impairment, stroke, and/or any neurological disorders like ADHD. All participants were paid an honorarium of $30 cash for their participation. The data reported here were collected as part of a larger study and were approved by the University of Alberta Research Ethics Board, and all participants provided informed consent.

### 2.2. Data Collection

2.2.1. Behavioural Data Collection

All Participants were Administered the Following Tasks:

Participants completed the Sight Word Efficiency (SWE) subtest and the Pseudo-Word Decoding Efficiency (PDE) subtest of the Test of Word Reading Efficiency—1st Edition (TOWRE) [38]. Participants were also administered the Word Identification (WI) and Word Attack (WA) subtests of the Woodcock Reading Mastery Test—Revised Normative Update (WRMT—R NU; [39]) to assess real-word and non-word reading skills. Extracted measures included fluency (i.e., the number of words that an individual could accurately identify within 45 seconds) from the TOWRE subtests and accuracy (i.e., the number of points scored divided by number of points possible) from the WRMT—R NU subtests.

Spelling skills were assessed using the Wide Range Achievement Test—4th Edition (WRAT4; [40]) spelling subtest. This dictation-based subtest evaluates an individual's ability to identify sounds and transfer them into a written form, and it is commonly used to evaluate spelling in adults (see [23,41,42].

The Matrix Reasoning (MR) subtest from the Wechsler Abbreviated Scale of Intelligence (WASI; [43]) was used to assess participants' non-verbal intelligence. Participants were asked to look at pictures of shapes and either name or point to the correct answer when given five response options. Both participant groups underwent the nonverbal IQ test in order to have a comparable performance measure from both groups [3,6,8].

2.2.2. fMRI Data Collection

Neuroimaging Tasks:

Participants completed three conditions of the letter probe task. In the letter probe condition, participants were given the auditory presentation of either a word or a non-word (duration = 2000 ms), followed by the visual presentation of a single letter on the screen (duration = 2000 ms). They were asked to indicate if the letter was, or was not, in the spelling of the word that they just heard. The letter probe task has been previously used to study the neural activity for spelling [44,45]. Each condition (described below) was

presented separately, and the stimuli within each condition was randomized (i.e., in an event-related design) with 25 baseline/fixation trials (i.e., where participants did not make any response). The interstimulus interval ranged from 500 ms to 18 s. The presentation of the three conditions were randomized for all participants with specific instructions preceding each run. The three conditions were as follows:

(1) Orthographic (O) condition: This condition used words that have an irregular spelling-to-sound correspondence so the retrieval of the spelling of the words was necessary in order to make the judgment. The letter option that was given was either a) absent from the pronunciation of the word (e.g., 'T' in 'gourmet'), b) ambiguous with respect to associated phonemes (e.g., 'C' in 'cello'), or c) highly associated with a specific phoneme (e.g., 'G' associated with /g/, as in 'get') but was pronounced differently in a selected word (e.g., sound of 'G' in 'regime'). In each case, the decision of the letter probe could not be made by the sound or pronunciation of the words alone.

(2) Orthographic–Phonological (OP) condition: The words in this condition had consistent spelling-to-sound correspondence (e.g., letter 'A' in 'gaze'). Thus, participants could utilize sound-based information, pronunciation-based information, or print-based information.

(3) Phonological (P) condition: The stimuli in this condition were pseudowords (e.g., letter 'N' in 'bint'), for which there were no stored whole-word sound, pronunciation-based representations, or print-based representations to retrieve. Therefore, participants had to generate the spelling of these stimuli to make the decision of whether the letter was in the spelling of the word or not.

A total number of 75 words (referred to as stimuli hereafter) were selected for each condition (See Table S2). Stimuli were matched on the following characteristics across and within the tasks: written frequency, orthographic and phonological neighborhood size, number of phonemes, syllables and morphemes, word length, summed bigram frequency, and summed bigram frequency by position [46].

The audio files were recorded by a male talker of central Canadian English at a sampling rate of 48 KHz using an M-Audio recording device in a sound-treated room. Each file was segmented, preprocessed, and calibrated for level using the Audacity software and stored as a single wav file. The three tasks were programmed using the software EPrime 2.0 Professional (Psychology Software Tools, Inc.).

*2.3. Procedure*

Once consented, each participant completed the behavioural test battery, which included the four reading assessments and the spelling and non-verbal intelligence tasks noted above. Performance measures included accuracy and/or rate (correct items over total time) for each behavioural task. Then, participants were provided with an overview of the experimental tasks in the fMRI and completed a practice trial in the behavioural testing room prior to going into the MRI scanner. Participants then walked over to Peter S. Allen Research Centre with the research assistants. They were screened by the MR technician to ensure that it was safe for them to go into the MRI. Once in the MRI, and prior to each task, participants were reminded about the nature of the tasks they were to complete. The EPrime software (Psychology Software Tools, Inc., http://www.pstnet.com) was used to present the stimuli for each task onto a screen, which was visible to the participants through a mirror attached to the head coil. The three tasks were counterbalanced, and stimuli in each task were presented randomly without replacement. Response time was operationalized as the time from stimulus presentation to the button response provided by the participant.

Images were acquired on a 3T Siemens Sonata scanner and were positioned along the anterior–posterior-commissure line. Anatomical scans included a high-resolution axial T1 MPRAGE sequence with the following parameters: repetition time (TR) of 1980 ms, echo time (TE) of 2.21 ms, number of slices of 176, base resolution of $232 \times 256 \times 176$ with the voxel size of $1 \times 1 \times 1$ mm, and scan time of 4.50 min. For each condition (O, OP, and P), a separate, single run, event-related design, whereby the 75 task stimuli were

randomly presented throughout a single run, was employed. The auditory presentation of each stimulus lasted 2000 ms. Participants had 2000 ms to respond to the stimuli. The interstimulus interval ranged from 500 ms to 18 s. Each run consisted of 230 volumes of 64 slice, axial spin, echo planar images (EPIs), obtained with the following parameters: a TR of 1980 ms, a TE of 30 ms, a base resolution of $64 \times 64$ with a $128 \times 128$ reconstruction matrix that improved pixel resolution through zero-filling prior to Fourier transform reconstruction, and a scan time of approximately 8 minutes. EPI slice thickness was 2.2 mm with no gap between slices. For the resting state sequence, functional T2* images were acquired using an echo-planar imaging sequence with the following parameters: a 2.2 mm isotropic voxel size, 64 interleaved slices, TR = 1980 ms, TE = 30 ms, and 242 measurements. The functional resting-state condition was 6 minutes long, and participants were instructed to keep their eyes open and look at the crosshair on the screen.

*2.4. Analysis*

2.4.1. Behavioural Data Analysis

Accuracy rates were calculated for all behavioural tasks and were compared between adults with and without reading impairments using independent sample *t*-tests at an adjusted $p < 0.05$ (for the number of tests to minimize type 1 errors). In-scanner behavioural performance (i.e., accuracy rates and reaction times) was also calculated for all three conditions and compared between groups using independent sample *t*-tests at an adjusted $p < 0.05$ (for the number of tests to minimize type 1 errors).

2.4.2. Functional Connectivity Analyses

Functional connectivity analysis was performed with the CONN-fMRI toolbox (version 18.b) in SPM 12 [47]. First, the functional and structural data were subjected to the standard preprocessing analysis in the CONN program. This pipeline included the functional realignment of functional images to each other; slice-timing correction; the segmentation of functional and structural images into gray matter, white matter, and cerebrospinal fluid maps; the normalization of data into standard Montreal Neurological Imaging (MNI) space; and spatial smoothing using a Gaussian kernel (8 mm) and denoising. Effects of within-subject variables of realignment and scrubbing and main condition effects (i.e., effect of task conditions) were regressed out, and band-pass filtering was executed at 0.1 Hz and above. The realignment variable consisted of 12 estimated subject motion parameters. Scrubbing identified the number of outlier scans for each subject. These two variables and task effects were considered to be potential confounding effects, and they were removed separately for each voxel, each subject, and each functional run/session. Additional preprocessing steps included band-pass filtering (0.1 Hz and above) and the inclusion of estimated subject motion and signal of white matter and cerebrospinal fluid as covariates of no interest.

Motion correction was performed using the Artifact Detection Toolbox (ART) (integrated in CONN), which identified outliers in global signal intensity and motion in the functional data time series. These outliers were included as first-level covariates of no interest to regress out their effects. Additional noise correction was performed using the anatomical component-based noise correction method (aCompCor). Global signal regression (GSR) was not performed, as GSR has been known to introduce spurious anti-correlations [48–50]. Because aCompCor allows for the interpretation of anticorrelations, this component-based noise reduction method was favored instead of the GSR method. aCompCor models noise from white matter, CSF, and regresses out the signal from these noise regions of interest (ROIs) in the first-level general linear model [47]. There were no between-group differences on the mean and maximum motion during the task sessions or between the global signal change and the number of valid scans used in the analysis (FDR-corrected; $p < 0.05$).

### 2.4.3. Brain Regions of Interests

We defined a spelling-network (SpN) that included left-hemispheric areas associated with print (fusiform gyrus (FFG) and inferior temporal gyrus (ITG)), sound (superior temporal gyrus (STG) and supramarginal gyrus (SMG)), meaning (angular gyrus (AG) and middle temporal gyrus (MTG)), and articulatory processing (inferior frontal gyrus (IFG) and supplementary motor area (SMA)) [3,6,9,45,51–53] (See Figure 1a and Table S2 in Supplementary Materials for MNI coordinates). The regions were spherical and 6 mm in radius, and they were developed using the Mango program. In keeping with previous literature, we characterized the networks as a function of three seed regions of interest (i.e., sound (supramarginal gyrus (SMG)), print (fusiform gyrus (FFG)), and articulation (inferior frontal gyrus (IFG)), from which all connectivity maps were generated.

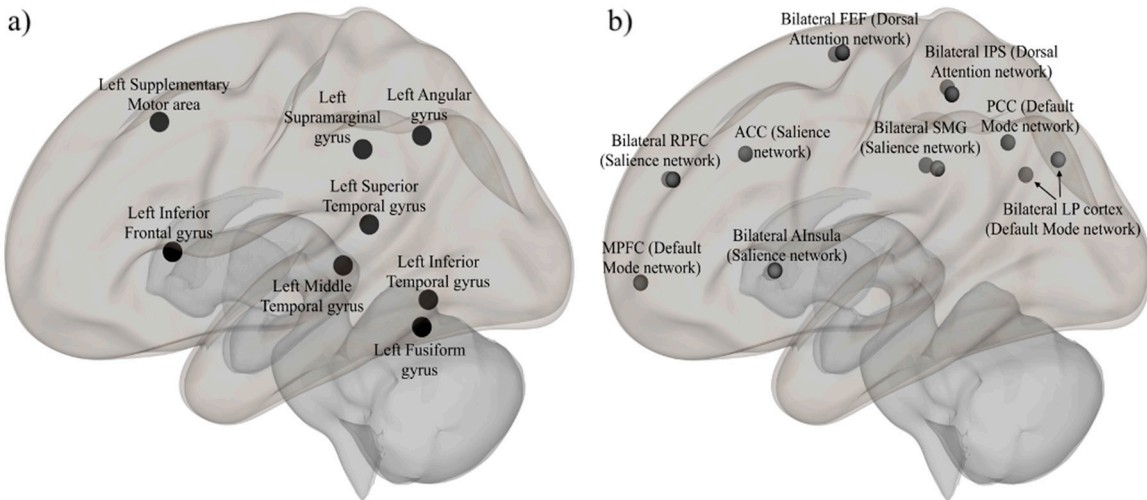

**Figure 1.** Regions of interest of the (**a**) spelling network (SpN) and the (**b**) resting-state networks (RSNs).

We conducted an ROI-to-ROI analysis where the functional connectivity was analyzed between a-priori selected pairs of ROIs. By using a defined set of ROIs, we could test every connection within the SpN between skilled and impaired readers without over testing the networks and inflating type 1 errors. The time series were computed and averaged across all voxels within the ROI, and the first-level correlation maps were made for each participant. These correlation maps were made by computing Pearson correlation between the residual BOLD time course of the seed region and the time course of all the other possible pairs of regions. These correlation coefficients were converted to normally distributed Fisher-transformed correlation coefficients to allow for second-level analyses at the group level.

For the resting-state networks (RSN), we selected the default mode, dorsal attention, and salience networks. These RSNs have been reliably and consistently detected in multiple studies [54–56]. ROIs for the resting-state networks include: the DMN (medial prefrontal cortex (MPFC), bilateral lateral parietal (LP), and posterior cingulate cortex (PCC)), the DAN (bilateral frontal eye field (FEF) and bilateral inferior parietal sulcus (IPS)), and the salience network (bilateral anterior insula (AInsula), bilateral rostral prefrontal cortex (RPFC), anterior cingulate cortex (ACC), and bilateral supramarginal gyrus (SMG)) (see Figure 1b). These network ROIs were a result of the independent component analysis of 497 individuals from the Human Connectome Project (HCP). All of the ROIs (for both SpN and RSNs) were delineated on a standardized MNI template to which each participant's structural and functional scans were aligned.

Functional connectivity at the group level involved performing a one-sample *t*-test of the Fisher values against zero. An independent sample *t*-test, with 1st-level connectivity maps, was conducted to examine the between-group differences in functional connectivity

from each of the three seed regions noted above. Analyses were separately run to determine any significant relationship between functional connectivity and spelling behaviour via a linear regression of connectivity and reaction time behaviour for each group. For the SpNs, we used a standardized spelling behaviour (outside of scanner). For the RSNs, we used the reaction time as acquired from the in-scanner spelling performance and the standardized spelling behaviour. The dependent variable was connectivity strength (i.e., Fisher-transformed z-scores) between the ROIs, and the independent variable was spelling behaviour.

## 3. Results

### 3.1. Behavioural Performance

Means and standard deviations for all behavioral tasks and the independent *t*-test results are summarized in Table 1. We found significant between-group differences on all the behavioural tasks except for nonverbal IQ. In general, the impaired group had lower accuracy rates and took longer to complete the tasks when compared to the skilled group. Between-group differences for in-scanner reaction times were found for all three conditions, such that the impaired group took longer to respond than the skilled group.

**Table 1.** Mean (standard deviation) and *p*-values (for independent sample *t*-tests) for all the behavioural measures. TOWRE: Test of Word Reading Efficiency—1st Edition; SWE: Sight Word Efficiency; PDE: Pseudo-Word Decoding Efficiency; O: orthographic condition; OP: orthographic–phonological condition; P: phonological condition. The asterisks indicate significant between-group differences, * $p < 0.05$. ** $p < 0.001$.

| | Skilled Group Mean (SD) | Impaired Group Mean (SD) | *t* Values | *p* Values |
|---|---|---|---|---|
| **Age (years)** | 21.58 (2.04) | 24.36 (5.36) | −2.08 | 0.46 |
| **Gender (% female)** | 73 | 71 | 0.14 | 0.89 |
| **TOWRE-SWE fluency (raw scores)** | 2.10 (0.26) | 1.81 (0.27) | 3.15 | 0.004 * |
| **TOWRE-SWE fluency (standardized scores)** | 96.63 (11.29) | 83.36 (8.78) | 3.59 | 0.001 * |
| **TOWRE-PDE fluency (raw scores)** | 1.36 (0.16) | 0.90 (0.23) | 6.73 | <0.001 ** |
| **TOWRE-PDE fluency (standardized scores)** | 105.53 (8.74) | 84.43 (9.42) | 6.63 | <0.001 ** |
| **Word identification** | 95 (0.05) | 79 (0.11) | 5.76 | <0.001 ** |
| **Word Attack** | 91 (0.07) | 72 (0.11) | 6.04 | <0.001 ** |
| **Spelling (raw score out of 42)** | 35.47 (2.89) | 28.29 (6.60) | 4.24 | <0.001 ** |
| **Spelling (standardized score)** | 84.58 (4.81) | 72.71 (10.22) | 4.45 | 0.001 * |
| **Non-verbal IQ** | 82 (0.07) | 81 (0.06) | 0.60 | 0.554 |
| **O condition** | | | | |
| **Reaction time** | 861.24 (120.81) | 980.09 (161.74) | −2.42 | 0.022 * |
| **Accuracy** | 0.78 (0.09) | 0.69 (0.14) | 2.33 | 0.052 |
| **OP condition** | | | | |
| **Reaction time** | 810.30 (125.17) | 927.22 (135.18) | −2.56 | 0.015 * |
| **Accuracy** | 0.86 (0.07) | 0.79 (0.11) | 2.40 | 0.046 * |
| **P condition** | | | | |
| **Reaction time** | 846.19 (125.53) | 946.85 (125.83) | −2.30 | 0.029 * |
| **Accuracy** | 0.83 (0.10) | 0.80 (0.13) | 1.15 | 0.337 |

### 3.2. Within-Network SpN Functional Connectivity

3.2.1. Characterize the Functional Connectivity of the Spelling Network (SpN) during the Three In-Task Conditions in People with and without Reading Impairments

To characterize the functional connectivity of the seed regions within the SpN, first-level ROI-to-ROI connectivity maps for each participant in each group were entered in a one-sample *t*-test, separately for each seed. Group-level connectivity maps (and associated statistics) are discussed for each seed region separately. Each analysis/map was FDR-corrected at the network level ($p < 0.05$). For the sake of brevity, and given the descriptive nature of this first research question, we only provide the visual representations for the O condition in the text, and a comprehensive list of results for the OP and P conditions is provided in the Supplementary Materials.

Inferior Frontal Gyrus (Speech)

Across the O (Figure 2i; Table 2), P, and OP conditions, for the skilled and impaired groups, IFG was connected to areas in frontal (SMA), parietal (SMG and AG), and temporal areas (STG, MTG, and FFG; See Tables S3 and S6 and Figures S1 and S4 for OP conditions in Supplementary Materials).

**Table 2.** Correlation/beta values, *t*-values and p-values for functional connectivity of the spelling network (SpN) for the Orthographic condition: (**i**) Inferior frontal gyrus, (**ii**) left fusiform gyrus and (**iii**) supramarginal gyrus. FDR-correction ($p < 0.05$).

| **(i) Inferior Frontal Gyrus** | | | | | | | |
| --- | --- | --- | --- | --- | --- | --- | --- |
| | **Skilled Group** | | | | **Impaired Group** | | |
| **Targets** | **Beta** | **t(18)** | ***p*-FDR** | **Targets** | **Beta** | **t(13)** | ***p*-FDR** |
| **SMA** | 0.27 | 5.44 | 0.000255 | **SMA** | 0.17 | 6.11 | 0.000262 |
| **MTG** | 0.17 | 3.6 | 0.007215 | **MTG** | 0.14 | 4.19 | 0.003689 |
| **SMG** | 0.11 | 2.94 | 0.013688 | **STG** | 0.14 | 3.12 | 0.018907 |
| **STG** | 0.11 | 2.91 | 0.013688 | **AG** | 0.13 | 2.48 | 0.046241 |
| **FFG** | 0.13 | 2.89 | 0.013688 | **FFG** | 0.09 | 2.38 | 0.046241 |
| **(ii) Left Fusiform Gyrus** | | | | | | | |
| | **Skilled Group** | | | | **Impaired Group** | | |
| **Targets** | **Beta** | **t(18)** | ***p*-FDR** | **Targets** | **Beta** | **t(13)** | ***p*-FDR** |
| **ITG** | 0.42 | 5.65 | 0.000164 | **ITG** | 0.39 | 10.91 | < 0.0001 |
| **SMG** | 0.15 | 4.15 | 0.0021 | **MTG** | 0.19 | 4.54 | 0.001948 |
| **IFG** | 0.13 | 2.89 | 0.019768 | | | | |
| **MTG** | 0.21 | 2.8 | 0.019768 | | | | |
| **STG** | 0.11 | 2.72 | 0.019768 | | | | |
| **SMA** | 0.21 | 2.37 | 0.034309 | | | | |
| **AG** | −0.14 | −2.05 | 0.054903 | | | | |
| **(iii) Supramarginal Gyrus** | | | | | | | |
| | **Skilled Group** | | | | **Impaired Group** | | |
| **Targets** | **Beta** | **t(18)** | ***p*-FDR** | **Targets** | **Beta** | **t(13)** | ***p*-FDR** |
| **FFG** | 0.15 | 4.15 | 0.0042 | **ITG** | 0.14 | 3.02 | 0.048354 |
| **SMA** | 0.14 | 3.73 | 0.005043 | **AG** | 0.16 | 2.84 | 0.048354 |
| **ITG** | 0.2 | 3.58 | 0.005043 | | | | |
| **IFG** | 0.11 | 2.94 | 0.015243 | | | | |

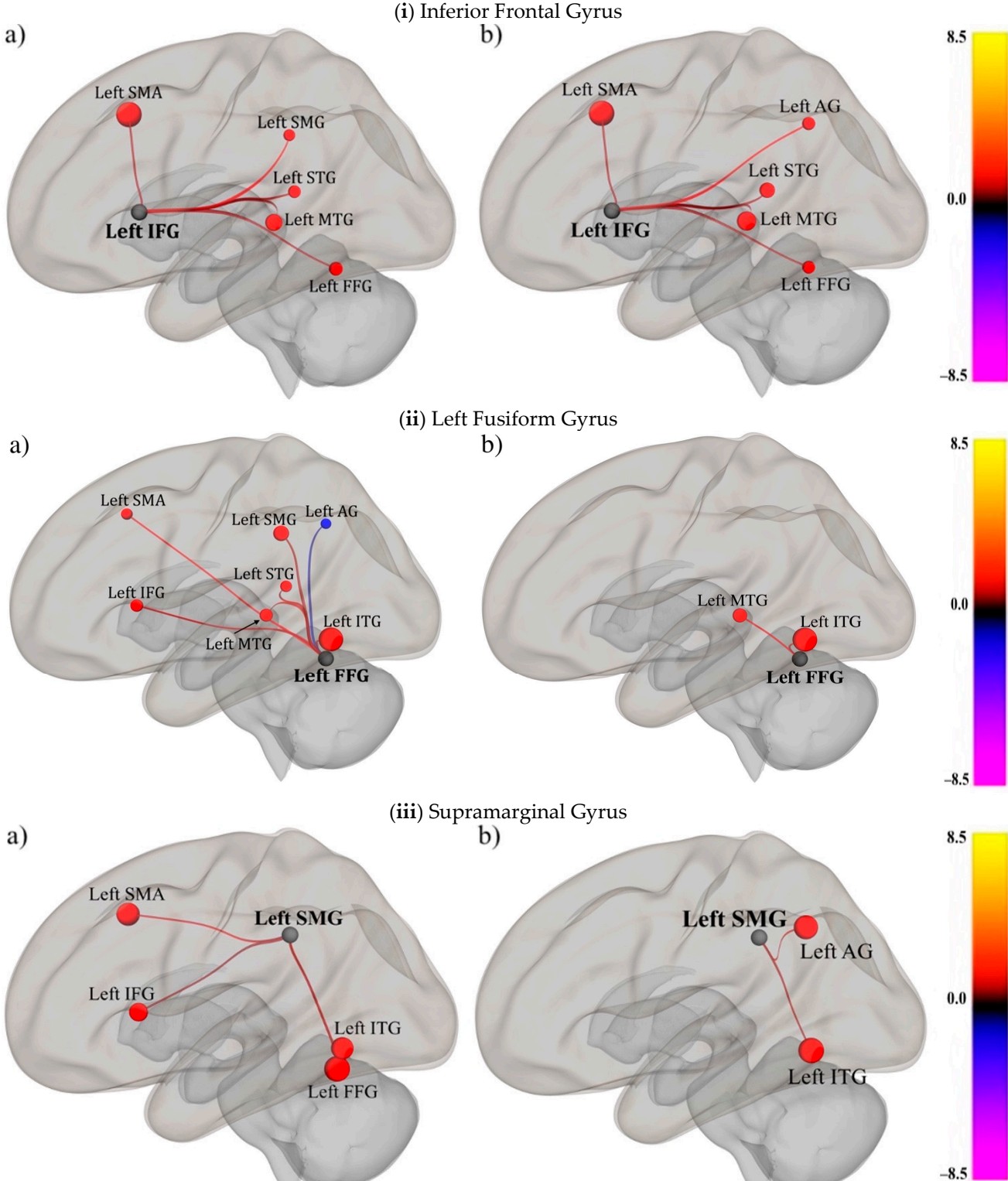

**Figure 2.** Functional connectivity within the SpN for the O condition in the (**a**) skilled group and the (**b**) impaired group, FDR-correction ($p < 0.05$) for the (**i**) inferior frontal gyrus, (**ii**) left fusiform gyrus, and (**iii**) supramarginal gyrus. The seed region is indicated in black, and the color-bar indicates the *t*-values. The size and the color of the target brain areas also indicate the *t*-values associated with the connectivity strength.

Fusiform Gyrus (Print)

In the O condition, FFG was significantly connected to all the target regions in the spelling network in the control group (i.e., seven connections), whereas the impaired group only had wo significant connections (with ITG and MTG) (Figure 2ii and Table 2).

For the OP and P conditions, the connectivity profiles between the two groups were comparable, with connections to frontal, parieto-temporal, and temporal regions (see Tables S4 and S7 and Figures S2 and S5 in Supplementary Materials).

Supramarginal Gyrus (Sound)

In the O, OP, and P conditions, SMG was significantly connected to multiple areas in the frontal (IFG and SMA) parieto-temporal (STG), and temporal regions (FFG and ITG) in both the control and impaired groups (see Tables S5 and S8 and Figures S3 and S6 in Supplementary Materials). The only deviation from this pattern was in the O condition for the impaired group, where there were no significant frontal connections with the SMG (see Figure 2iii and Table 2).

Independent Sample *t*-Tests:

Next, we investigated whether there were significant between-group differences in connectivity from the three seed regions within the SpN. If we took an average of the standard deviations of the r-values from each group (approximately 0.14), an alpha at 0.05, and power = 0.80, we would need a minimum difference of 0.15 between the groups to detect a significant effect. For each of the seed regions (i.e., IFG, FFG, and SMG), first-level ROI-to-ROI connectivity maps for each participant were entered into an independent sample *t*-test. No significant effects survived the stringent FDR corrections. Thus, in an attempt to mitigate type 2 errors, we investigated a more liberal corrected threshold ($p < 0.01$). Again, there were no surviving significant connections.

3.2.2. Examine the Relationships between Domain-Specific Networks (i.e., SpN) and Domain-General RSNs in People with and without Literacy Impairments

To characterize the functional connectivity between the SpN seed regions and RSNs, level ROI-to-ROI connectivity maps for each participant in each group were first entered into a one-sample *t*-test, separately for each seed. Group-level connectivity maps (and associated statistics) are discussed for each seed region separately. Each analysis/map was FDR-corrected at the network level ($p < 0.05$). For the sake of brevity, we only provide the visual representations for the O condition in the text; however, the results of the OP and P conditions are fully presented in the Supplementary Materials.

Inferior Frontal Gyrus (Speech)

For the skilled group, in each of the O, OP, and P conditions, the IFG was significantly connected with ACC and bilateral AInsula (Figure 3i and Table 3 for connectivity for O condition).

In contrast, for the impaired group, in each of the O, OP, and P conditions, the IFG was connected to multiple salience (bilateral AInsula, ACC, SMG, and RPFC) and dorsal attention network areas (bilateral IPS) across the three conditions (See Tables S9 and S12 and Figures S7 and S10 in Supplementary Materials).

Fusiform Gyrus (Print)

Across the three conditions, FFG had significant connections to multiple RSNs, including the DAN (bilateral FEF), the DMN (MPFC), and the salience network (bilateral AInsula, ACC, and IPS) in the skilled group (see Figure 3ii for connectivity for the O condition).

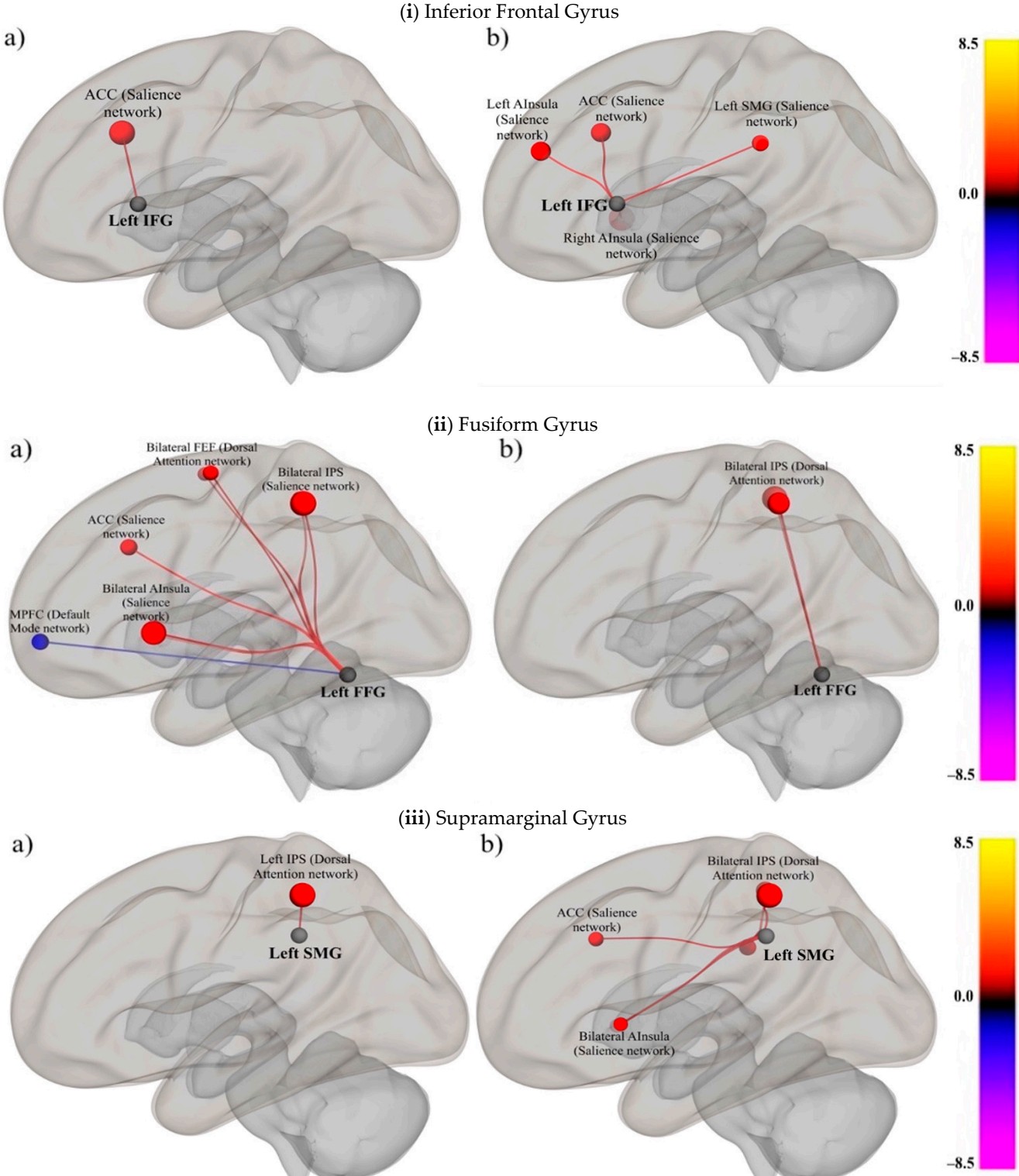

**Figure 3.** Functional connectivity of the RSNs with the SpN network for the Orthographic condition in the (**a**) skilled and (**b**) impaired groups: (**i**) inferior frontal gyrus, (**ii**) fusiform gyrus, and (**iii**) supramarginal gyrus. FDR-correction ($p < 0.05$). The seed region is indicated in black, and the color-bar indicates the $t$-values. The size and the color of the target brain areas also indicate the $t$-values associated with the connectivity strength.

**Table 3.** Correlation/beta values, *t*-values and *p*-values for functional connectivity between the resting-state networks (RSN) and spelling-networks (SpNs) of (**i**) IFG, (**ii**) FFG, and (**iii**) SMG for the O condition for skilled and impaired groups. FDR-correction (*p* < 0.05). ACC: anterior cingulate cortex; AInsula: Anterior Insula; RPFC: rostral prefrontal cortex; IPS: inferior parietal sulcus; FEF: frontal eye field; MPFC: medial prefrontal cortex.

| (**i**) Inferior Frontal Gyrus | | | | | | | |
| --- | --- | --- | --- | --- | --- | --- | --- |
| **Skilled Group** | | | | **Impaired Group** | | | |
| **Targets** | **Beta** | **t(18)** | ***p*-FDR** | **Targets** | **Beta** | **t(13)** | ***p*-FDR** |
| **ACC** | 0.22 | 3.82 | 0.017542 | **Right AInsula** | 0.27 | 6.02 | 0.000601 |
| | | | | **ACC** | 0.22 | 5.53 | 0.000682 |
| | | | | **Left SMG** | 0.21 | 4.96 | 0.001218 |
| | | | | **Left RPFC** | 0.17 | 4.58 | 0.001806 |
| | | | | **Left IPS** | 0.15 | 2.72 | 0.049002 |
| (**ii**) Fusiform Gyrus | | | | | | | |
| **Skilled Group** | | | | **Impaired Group** | | | |
| **Targets** | **Beta** | **t(18)** | ***p*-FDR** | **Targets** | **Beta** | **t(13)** | ***p*-FDR** |
| **Left IPS** | 0.26 | 5.82 | 0.000225 | **Right IPS** | 0.20 | 6.62 | 0.000248 |
| **Left AInsula** | 0.15 | 5.53 | 0.000225 | **Left IPS** | 0.23 | 5.54 | 0.000719 |
| **MPFC** | −0.13 | 3.52 | 0.009241 | | | | |
| **ACC** | 0.16 | 3.52 | 0.009241 | | | | |
| **Left FEF** | 0.15 | 3.13 | 0.017402 | | | | |
| **Right AInsula** | 0.09 | 2.83 | 0.022545 | | | | |
| **Right IPS** | 0.14 | 2.81 | 0.022545 | | | | |
| **Right FEF** | 0.14 | 2.79 | 0.022545 | | | | |
| (**iii**) Supramarginal Gyrus | | | | | | | |
| **Skilled Group** | | | | **Impaired Group** | | | |
| **Targets** | **Beta** | **t(18)** | ***p*-FDR** | **Targets** | **Beta** | **t(13)** | ***p*-FDR** |
| **Left IPS** | 0.26 | 6.04 | 0.000146 | **Left IPS** | 0.31 | 6.92 | 0.000147 |
| | | | | **Right IPS** | 0.21 | 4.52 | 0.003995 |
| | | | | **Right SMG** | 0.18 | 3.87 | 0.009018 |
| | | | | **ACC** | 0.15 | 3.29 | 0.020636 |
| | | | | **Right AInsula** | 0.15 | 3.07 | 0.023493 |
| | | | | **Left AInsula** | 0.21 | 3.01 | 0.023493 |
| | | | | **Left RPFC** | 0.14 | 2.52 | 0.050879 |

For the impaired group, the highest number of surviving connections were present in the OP condition, with areas like the bilateral IPS, bilateral FEF, ACC, AInsula, and PCC being connected. However, FFG was connected to fewer areas in the O (Figure 3iii and Table 3) and P conditions, with surviving connections to bilateral IPS and AInsula (See Tables S10 and S13 and Figures S8 and S11 in Supplementary Materials).

Supramarginal Gyrus (Sound)

In the skilled group, SMG was connected to bilateral IPS and AInsula in the O and OP conditions, with additional connections to MPFC and RPFC in the P condition.

On the other hand, SMG had significant connections with multiple areas in the RSNs of the salience (AInsula, ACC, bilateral RPFC, and SMG) and DAN (bilateral

IPS) networks in the impaired group (See Tables S11 and S14 and Figures S9 and S12 in Supplementary Materials).

Independent Sample *t*-Tests

No significant (FDR-corrected at the network level) between-group differences of the SpN–RSN connections, from any of the seed regions, emerged. We followed-up with the looking at the pairwise connectivity levels at a liberal corrected $p < 0.01$ threshold to provide a preliminary characterization of the nature of connectivity in the control and impaired groups.

Significant findings are described next.

Inferior Frontal Gyrus (Speech)

There were between-group differences in how the IFG was connected with the DAN IPS ($t = -2.70$, $p = 0.01$, and Cohen's d = 0.13) in the P condition, such that the impaired group was more positively connected (mean = 0.13 and SD = 0.14) than the control group (mean = 0.01 and SD = 0.12).

Fusiform Gyrus (Print)

In the OP condition, there was a between-group difference in how the FFG was connected with the DMN lateral parietal ($t = -3.09$, $p = 0.004$, and Cohen's d = 0.16), with the FFG being negatively connected to lateral parietal in the control group (mean = $-0.02$ and SD = 0.13) while being positively connected in the impaired group (mean = 0.14 and SD = 0.19). Additionally, FFG was also differently connected to the DMN PCC ($t = -2.68$, $p = 0.012$, and Cohen's d = 0.12). The mean connectivity strength of the FFG–PCC was significantly lower in the control group (mean = $4.1 \times 10^{-4}$ and SD = 0.11) than the impaired group (mean = 0.11 and SD = 0.13).

3.2.3. Examine the Relationships between SpN Connectivity and Spelling Behaviour in People with and without Reading Impairments

To investigate connectivity–behaviour relationships for each group, a linear regression of connectivity strength was run. The dependent variable was connectivity strength between the ROIs, and the independent variable was the standardized outside of scanner spelling behaviour.

There were no significant (FDR-corrected or liberal-corrected $p < 0.01$) correlations between SpN connectivity and spelling behaviour in either group.

3.2.4. Examine the Relationships between RSN Connectivity and Spelling Behaviour in People with and without Literacy Impairments

We investigated the connectivity–behaviour relationship between the RSNs and outside-of-scanner spelling score (standardized) for both groups. Each analysis was FDR-corrected at the analysis level ($p < 0.05$).

There were no significant (FDR-corrected or liberal-corrected $p < 0.01$) correlations between SpN connectivity and spelling behaviour in either group.

We also followed-up with a less stringent corrected $p < 0.01$ thresholds to assess how connectivity related to spelling behaviour in the skilled and impaired group.

Fusiform Gyrus (Print)

The left FFG–left SMG (salience network) was negatively related to spelling accuracy score (from the OP condition) in the control group (r = $-0.62$ and $p = 0.05$), but the impaired group did not show any significant relationship (r = $-0.141$ and $p = 0.631$) (Figure S13 in Supplementary Materials).

Inferior Frontal Gyrus (Speech)

The IFG–MPFC (DMN) connectivity was positively related to the OP reaction time in the control group ($r = 0.57$ and $p = 0.01$), while the impaired group did not show any significant relationship ($r = -0.42$ and $p = 0.13$) (see Figure S14 in Supplementary Materials).

## 4. Discussion

The current paper aimed to address the current gaps in our understanding of the spelling network by (1) characterizing the domain-specific spelling networks of skilled and literacy-impaired individuals during a spelling task, (2) determining the extent to which the domain-specific network is related to domain-general (e.g., attention-specific) brain networks, (3) quantifying how the domain-specific spelling network is related to spelling performance, and, finally, (4) assessing the extent to which domain-general brain networks are related to spelling behaviour. Two main findings emerged. First, the SpNs for individuals with literacy impairments were tightly coupled to their RSNs. The same was not true for individuals without literacy impairments. In contrast, the RSNs for skilled reading individuals predicted their spelling behaviour. However, such relationships were not evident for individuals with literacy impairments. We address the implications of these findings to understand the role of sound, print, and articulation in written communication and discuss how these findings inform our understanding of spelling performance in individuals with and without literacy impairments.

### 4.1. Characterization of the Spelling Network (SpN)

Connectivity during Retrieval of Whole-Word Representations in the Impaired Group

Overall, we did not find any statistically significant differences in the strength of connections between groups with respect to the SpN networks. However, it is worthwhile discussing the SpNs from a descriptive perspective because there is little information in the literature on spelling networks in general. First, skilled individuals had highly connected networks in each of the spelling conditions (O, OP, and P), with 16, 17, and 17 connections surviving the threshold, respectively. Such high connectivity speaks to the relatively complex nature of the spelling process, namely the integration of print, sound, and articulatory representations for successful decoding [57]. The individuals with literacy impairments had comparatively fewer connections that survived the statistical threshold specific to the O condition (a total of nine significant connections) vs. the P and OP conditions (20 and 14, respectively). Furthermore, it was the FFG (print processing; two significant connections) and SMG (sound-to-letter mapping; two significant connections) seed regions that had minimal significant connections. These results were in line with previous studies that have found reduced structural [8,58–60] and functional connectivity in individuals with reading impairments [3,6,9]. When we spell, we need to access and integrate the phonological (sound of words), semantic (the meaning of the word), orthographic (print), and articulatory (sounding out the word) units of information in an efficient manner to correctly recover and generate accurate spelling representations [57]. That this pattern of findings was notable in the FFG, a key region involved in the fast identification and mapping of orthographic representations, and during the orthographic condition, provides some evidence for the aberrant functioning of this portion of the network [3,15,16]. Similarly, the SMG, also had only two connections in the O condition, which may be indicative of a disrupted multimodal integration between sound (of the word) and print (of the letter). While the connection strengths within the SpN were not significantly different between the skilled and impaired readers, the more global reduction in connections could potentially contribute to slower processing speed during spelling tasks, less compensatory mechanisms available to counteract aberrant or deficient connections, and/or fewer places where information can be double-checked for accuracy. The extent to which the inherent 'sub-threshold' pattern of SpN connectivity contributes to any one of (or all of) these outcomes is necessary to establish in future work.

*4.2. Connections between SpN and Domain-General RSNs*

4.2.1. Increased SpN–DMN Connections in Impaired Readers

Areas of the DMN were found to be differently connected with the SpN seed regions in the impaired group compared with the skilled group. Notably, the impaired group showed positive connections between the SpN nodes (FFG) and DMN regions (LP and PCC), which were stronger than any connections in the skilled networks. This was contrary to previous work on typical readers that has reported a negative relationship between task-negative DMN regions and task-positive regions [29,61]. It has been argued that the DMNs play a role in mind wandering, internal mentation, autobiographical memory, remembering past events, and planning future events that needs to be suppressed to reallocate resources to the task at hand. For example, the authors of [62] found that the stronger the negative relationship between the DMN and the attention-positive network, the less attentional lapses happened. Resting-state studies in the reading literature have also supported the presence of such an anti-correlated relationship, with the authors of [31] presenting evidence for negative connections between FFG/VWFA and DMN regions in skilled adult readers. However, we found that the typical competing relationship between task-positive and task-negative regions (as one goes up the other goes down) was not present for impaired reading individuals. In fact, individuals with reading impairments had comparatively more SpN–RSN connections (a total of 53 significant connections) than the skilled readers (36 significant connections). This descriptive finding, in conjunction with the statistically significant differences between the groups, leads us to speculate that, in people with a history of literacy impairments, the brain is not able to disengage from internal thinking to focus on the task at-hand, namely spelling. To our knowledge, there have been no studies that have found this kind of positive connection between DMN areas and task-positive regions in atypical reading individuals. Ultimately, much more work is needed to determine how the SpN–DMN relationships reported here and in previous work actually contribute to behavioural performance. That is, is it the general strength of the DMN connections (IFG–MPFC, FFG–MPFC, and LP regions) that contribute to improved literacy performance, or it is the efficiency at switching between the DMN and SpN networks that results in greater performance? Understanding these dynamic relationships is important for the advancement of remediation approaches for individuals with literacy challenges.

4.2.2. Altered SpN–Attention Connections

The inferior parietal sulcus (i.e., the DAN) was more positively connected with inferior frontal gyrus in the impaired group compared to the skilled group during the spelling generation condition (i.e., P task). As the DAN is involved in goal-directed top-down processing and selective attention, this points towards the increased need for attention in people with limited literacy in demanding conditions (generating nonwords in this example). Unfortunately, we did not obtain direct measures of attentional control in the current study (although we screened for ADHD) and thus cannot fully disentangle the extent to which these differences are a by-product of attention; however, the behavioural accuracy for the P condition was comparable between the groups, so we hypothesize that the increased connection may reflect a successful strategy employing selective attention. Nonetheless, this increased connectivity does help us in further characterizing the neurobiology of dyslexia in adulthood. It would be interesting to study the developmental trajectory of this connectivity with attention from pre-reading stages to skilled reading and remediation work that target attention and attention switching may be one avenue of support.

*4.3. SpN– and RSN–Behaviour Correlations.*

We did not find any relationships between SpNs and spelling behaviour for either group. While this is consistent with some work (see [20,21] papers for similar null findings), other studies have reported significant brain–behaviour relationships [63,64]. These in-

consistent null/positive effects in the brain-spelling domain are perplexing. The extent to which such reports are driven by under-powered studies, insensitive spelling measures, and/or a heterogeneous sample could not be determined with the current study. It is interesting that the reading domain does not suffer the same fate, which indicates that these null findings are somewhat specific to the spelling process itself. Is this because spelling is a more cognitively demanding task than reading? Perhaps the spelling network is not differentially engaged (i.e., whereby we might see differences in task demands); instead all the regions are globally involved. That is, in trying to disentangle unique contributions of specific regions, the inclusion of additional regions in the statistical model served to remove too much shared variance. While definitive conclusions about differential connections are premature, our findings do underscore the need for future work that examines SpN–behaviour connections. Perhaps SpN–behaviour subsequent to remediation would provide clarity, whereby malleable connections may provide additional insight, and then remediation approaches that target processes that influence such connections may be an ideal avenue to support adults with literacy impairments.

### 4.4. Limitations

While the results obtained in the present study were in line with the previous results, we do recognize our small sample size and the possibility that some of the null findings were a result of low power. In turn, the significant reported findings that are associated with liberally corrected $p$-values should also be interpreted with caution (i.e., may represent type 1 errors) and must be replicated before much can be said about the contribution of connectivity networks to spelling behaviour.

### 5. Conclusions

Here, we provide a neuropsychological profile of spelling in skilled and impaired readers. Using functional connectivity, we characterized the spelling networks across multiple tasks, across left and right hemisphere brain regions, and across varying levels of literacy performance. Overall, we found a highly connected network of regions during multiple spelling tasks for skilled readers, with a particular emphasis on the role of the IFG. In contrast, we report under connected profiles for individuals with literacy impairments, primarily in regions associated with print (e.g., FFG and ITG) and sound (e.g., SMG). These results provide evidence for the underlying connectivity patterns associated with spelling performance and also provide much-needed information for the advancement of theoretical models and remediation approaches. As individuals with reading impairments face lifelong issues with spelling performance, our hope is that the current work will stimulate further investigations into better understanding written language.

**Supplementary Materials:** The following are available online at https://www.mdpi.com/2673-4087/2/1/5/s1.

**Author Contributions:** Conceptualization, K.C. and J.C.; methodology, K.C. software, K.C.; validation, J.C. and W.E.H.; formal analysis, K.C. and J.C.; investigation. J.C.; resources, W.E.H.; data curation, K.C.; writing—original draft preparation, K.C., J.C. and W.E.H.; writing—review and editing, K.C., J.C., and W.E.H.; visualization, K.C.; supervision, J.C. and W.E.H.; project administration, J.C. and W.E.H.; funding acquisition, J.C. All authors have read and agreed to the published version of the manuscript.

**Funding:** This work was partially supported by the Natural Sciences and Engineering Research Council of Canada (NSERC) in the form of a scholarship to author KC and a grant to JC (386617-2012).

**Institutional Review Board Statement:** The study was conducted according to the guidelines of the Declaration of Helsinki and approved by the Institutional Review Board (or Ethics Committee) of University of Alberta (protocol code Pro00066347 and date of approval: 18 August 2016).

**Informed Consent Statement:** Informed consent was obtained from all subjects involved in the study.

**Data Availability Statement:** Data is contained within the article or supplementary material.

**Conflicts of Interest:** The authors declare no conflict of interest.

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
