# Peer review of "Is the Letter ‘t’ in the Word ‘gourmet’? Disruption in Task-Evoked Connectivity Networks in Adults with Impaired Literacy Skills"

_neurosci, doi:10.3390/neurosci2010005_

Round 1
Reviewer 1 Report
Review: Is the letter ‘t’ in the word ‘gourmet’? Disruption in task-evoked connectivity networks in adults with impaired literacy skills. The authors present an interesting work which focuses on quantifying differences in spelling task/network functional connectivity between a neurotypical and a group with impaired literacy. Although this is a very interesting topic, there were some concerns. Overall, the methods/analyses need to be described in more detail. The lack of a clear differences between groups is problematic and doesn’t match how these data are interpreted (e.g. one can’t say there is “differential connectivity” between groups when the authors don’t actually show this). There are also some concerns about the interpretation and logic behind the brain-behavior analysis. It's not sure the beginning of the title really captures the focus of this work. Overall, this work needs considerable revision.
Major/Minor issues in order of the paper:
1) On Page 5 line 227, it’s not clear what this description means. It sounds like it means that there were 230 volumes for each condition. Typically, one would state how many trials are associated with each condition, not the number of volumes. The number of volumes for each condition would be contingent on the duration of the trial and the associated BOLD response, and is much easier to consider with a blocked design.
“For each condition (O, OP and P), 230 volumes of 64 slice, axial spin, echo planar images (EPIs) were obtained with the following parameters: TR 1980 ms, TE 30 ms, base resolution 64 x 64 with a 128 x 128 reconstruction matrix that improved pixel resolution through zero-filling prior to Fourier transform reconstruction, scan time approximately 8 minutes.”
It is confusing because the authors claim there are 75 stimuli for each of the O, OP, and P conditions (page 5, line 199), but it’s not clear how these are spread out across the run or multiple runs.
2) How many LPT runs were there? Just one? I don’t know how to match the number of condition trials to the number of runs and I don’t know how they are spread across the run(s).
3) What is the timing of each trial? How long was the auditory word presented, how long was the letter presented for, and how much time did they have to respond? A figure might be helpful.
4) Was this a block or event-related design? Was the condition order alternating or psueodorandomized? If so, how?
5) how many baseline/fixation time points were there? Was the timing of the trials jittered? Or was there an equal amount of fixation timepoints between each trial?
6) What was the task based connectivity analysis run? Was it a psychophysiological interaction analysis?
7) What part of the trial was included in your condition regressors? Was it the entire trial or just the letter/response portion?
8) Did you account for inaccurate trials? Were there any no-response trials? Were these ignored?
8) In Table 2, how can the Impaired group have a 0 p-value for the ITG connection?
9) In section 3.2 it’s not clear why some conditions are discussed and others aren’t. E.g. for the Inferior frontal gyrus section there are vaguely described results for the O, P, and OP conditions (a better summary here would be recommended). Then in the next fusiform gyrus section it only discusses the O condition. Why isn’t there a mention of the P and OP conditions for the Fusiform gyrus (or Supramarginal gyrus section for that matter)? It needs to be made clearer what analyses were run and what analyses had significant results, and why some results are discussed in the main text (e.g. they directly relate to a hypothesis) and why some are not.
10) It’s not clear what to make of the results in section 3.2 given there is no significant difference between the groups. It’s not fair to claim in your discussion that there is “differential connectivity between the skilled and impaired readers” because you don’t actually detect a difference between them. It is likely you would see a difference with more subjects or with a different analysis, but not as presented. You can claim that the connectivity profile is say quantitatively lower in the Impaired group though. This is not a strong statement, but it does reflect the fact that although statistically there is no difference, the direction of the non-sig effect is consistent with your hypothesis and it also suggests that with more power you would observe a significant effect.
11) In results section C, why is the title “Examine the relationships between SpN connectivity and reading behavior…” isn’t the analysis focused on spelling behavior?
12) In Figure 3, I don’t understand the logic of the finding. So the faster a subject is at responding to the LPT, the less the connection between left IFG-ITG in the Impaired Group? Wouldn’t you expect that the stronger the connection the better (faster) the performance?
13) In your discussion on page 12, line 483 you claim “However, it was clear that the skilled participants had vastly more connections than the impaired individuals, particularly associated with the FFG and SMG seed regions.” This is a fine statement, but it isn’t clear why you didn’t try to quantify this, if this is actually the measure of interested used to discuss distinguishing the groups. There are plenty of graph-theoretical measures available in CONN that you can use to examine a network as a whole or the status of a node. E.g. you could measure the network measure “degree” or “cost” of each of the nodes in the groups.
14) Using the LPT reaction time as your behavioral measure is not ideal. Generally one can observe connectivity or activation differences just if someone responses faster/slower to a task. This may or may not be related to the condition or spelling per se, but instead to fluctuations in attention, for instance. So, first, it would be more impressive to quantify connectivity strength in relation to measures outside of the scanner (e.g. spelling), and second it is actually worth considering accounting for reaction time in your model either at the trial-by-trial level or at the group level (e.g., Yarkoni, Barch, Gray, Conturo, Braver, 2009; Barber, Caffo, Pekar, Mostofsky, 2018). In this manner you can account for and ignore, OR look at reaction time modulation effects more directly (if this argument is convincingly made by the authors that this is a worthwhile analysis to run...this is not immediatley clear to me).
15) A final concern regarding the reaction time is that you ran so many different analyses, that there is a concern that you are finding brain-behavioral results due to chance when spread across all of your different analyses (i.e. for each condition and each ROI of interest...9 in total?... was there a correction for multiple comparisons at this level?). If you revise, I would recommend discussing how you mesh the many different analyses you ran with the relevance/significance of only the results you present/highlight.
Reviewer 2 Report
The manuscript by Cheema, Hodgetts, and Cummine is a relatively comprehensive research article on the potential effects of the regional brain activity on some language functions. The authors focus on the domain specific networks and attempt to utilize brain imaging methods for the understanding of spelling. The experiments have been well designed and data has been well collected. For writing, the background has been well introduced in the paper.
There are some moderate concerns:
- It needs to be further discussed on the “domain specific spelling networks”(Line 133) considering other studies for spelling errors previously published.
- In section Materials and Methods 2.1 Participants. Line 141-145, it was unclear whether cognition tests included in the pretest.
- Lines 150-151, “neurological disorders like ADHD”, the psychiatric history of each participate should be considered if possible.
- In Table 1 and related results, the authors should provide F factors in addition to p value.
- Discussion section, Lines 466-467, the authors tested “spelling network” and “spelling performance”; it is unclear whether it could be developed for qualitative measures.
- Missing discussions on mechanisms of neural circuits and their changes. Missing information on the correlated brain regions for Lines 527-529.
- It was quite understood that the limited samples were obtained. The reviewer would suggest a recommendation if the authors could provide power analysis for all the tests.
Reviewer 3 Report
This paper investigated the functional connectivity network related to spelling and compared two groups of adults with/without weak literacy performance on connectivity in this network. Furthermore, connections with other circuitries associated with attention and salience were investigated. Results indicated a widespread pattern of differences between groups.
This is a very interesting and timely study. There is indeed only very little literature specifically on fMRI evidence on spelling and spelling deficits. Therefore, this paper targets research questions that are as far mostly unanswered.
There are a number of methodological issues though.
- First of all, it is unclear how participants were recruited. The criterion of performance > 1.5 SD below the skilled group in one reading task does not reflect standard clinical practice. Participant selection should be based on the norms of standardized tests. Furthermore, the selection criterion was based on reading, not spelling performance, therefore it does not seem to fit the aims of the study (“the overall goal of this paper is to characterize the brain networks for spelling processes”). Any difference in functional networks between groups could be attributed to reading rather than spelling problems, or a mixture of the two.
- Table 1 shows mean group performance and spelling is also reported. However, the reported mean values for the two groups of 84 and .67 do not seem to be on the same scale. Related to this point, are these raw or standardized values? I would appreciate to see standardized values in order to check whether the deficit group performed considerably below the typical group. Moreover, how many participants in the deficit group had spelling performance in the age-adequate range?
- As the authors pointed out, attention is strongly related to literacy and especially spelling performance. However, they did not measure attention skills behaviorally and therefore we do not know whether groups were comparable on this dimension. This is an importance confounding aspect.
- To investigate the associations between functional connectivity and behavior, the authors run correlations with the reaction times of the in-scanner task. This seems somehow circular to me, given that functional connectivity was elicited by the in-scanner task. Moreover, the in-scanner task most likely resembles processes that are recruited during spelling, including phonological, orthographic and working memory. However, given that the authors measured also spelling performance directly, I do not understand why they did not correlate behavioral spelling performance with functional activity.
- To address the association of functional connectivity with behavior, the authors performed a regression on “connectivity strength” between the ROIs. I may have missed it, but what is meant with this term statistically?
Minor points
- The abstract would improve in scientific quality if it presented a clear research question.
- In the introduction, the authors report a wealth of research findings. However, the cited evidence is not always extremely up-to-date. I would suggest to mention recent studies that investigated the functional correlates of isolated spelling deficits (https://www.tandfonline.com/doi/full/10.1080/23273798.2020.1859569), whose findings are only partly in line with previous evidence, as well as studies that addressed the association between reading and spelling performance with brain activity (see for example https://www.sciencedirect.com/science/article/pii/S1878929319300544 ).
- How was the effect size calculated for t-tests? I assume this was Cohen’s d, but it is not specified.
Round 2
Reviewer 3 Report
The authors responded to my queries. I believe the manuscript improved in quality and I do not have further comments.